# Empowering Local Practitioners to Collect and Report on Anthropogenic Riverine and Marine Debris Using Inexpensive Methods in India

Katharine A. Owens [1,*], Jaya Divakaran Sarasamma [2], Katie Conlon [3], Solomon Kiruba [4], Alwyn Biju [2], Niyathi Vijay [2], Manikandan Subramanian [2], Smitha Asok Vijayamma [5], Ayona Jayadev [5], Vineeta Hoon [6], Rebekah Padgett [6], Pranoti Joshi Khanolkar [7], Dilip K. Kakavipure [8], P. M. Mohan [9], Sourav Chattopadhyay [10] and Chinmay Khanolkar [11]

1. Department of Politics, Economics, and International Studies, University of Hartford, West Hartford, CT 06112, USA
2. Department of Environmental Sciences, University of Kerala, Thiruvananthapuram 695581, Kerala, India; jayvijayds@gmail.com (J.D.S.); alwynbiju@gmail.com (A.B.); niyathivijayan01@gmail.com (N.V.); manikandan@keralauniversity.ac.in (M.S.)
3. Toulan School of Urban Studies, Portland State University, Portland, OR 97201, USA; conlon@pdx.edu
4. Department of Zoology, Madras Christian College, Chennai 600059, Tamil Nadu, India; solomonkiruba@gmail.com
5. Department of Environmental Sciences, All Saints' College, Thiruvananthapuram 695007, Kerala, India; smithaasok@gmail.com (S.A.V.); sureshayona@gmail.com (A.J.)
6. Centre for Action Research on Environment Science and Society (CARESS), Chennai 600094, Tamil Nadu, India; caress.trust@gmail.com (V.H.); rpadgett7@gmail.com (R.P.)
7. Department of Science and Humanities, K. J. Somaiya College of Engineering, Mumbai 400077, Maharashtra, India; pranoti@somaiya.edu
8. Department of Zoology, B.N.N. College, Bhiwandi 421302, Maharashtra, India; dlpkakavipure@gmail.com
9. Department of Ocean Studies and Marine Biology, Pondicherry University Off Campus, Brookshabad, Port Blair 605014, Andamans, India; pmmnpu@rediffmail.com
10. Center of Advanced Study in Marine Biology, Annamalai University, Parangipettai 608502, Tamil Nadu, India; babutheoneonly@gmail.com
11. K. J. Somaiya College of Science and Commerce, Somaiya Vidyavihar University, Mumbai 400077, Maharashtra, India; c.khanolkar@somaiya.edu
* Correspondence: kowens@hartford.edu

**Abstract:** This article includes a review of the literature on marine debris in an Indian context and introduces a replicable, scientific, and inexpensive collection method to build capacity and inform policymakers. We share baseline data resulting from ten cleanups using these methods in India. This method was introduced in a 2019 workshop to train Indian researchers, leading to local-led collections in three states and two Union Territories (8 beaches, 2 riversides) yielding 33,474 individual pieces of debris weighing a total of 599.15 kg. Plastic was the most frequently found material at all ten collection sites, comprising from 45% to 89% of all items found. The research establishes a baseline data collection at ten locations, with debris density at sites ranging from 0.38–3.86 items/m². Application of the Clean Coast Index yields resulting rankings of moderate (1 site), dirty (2 sites), and extremely dirty (7 sites). Researchers also identified 2461 brands in analysis at six sites, 76% of which were Indian in origin. Replication of the methods in other Indian regions among the community of thirty-three practitioners was below target for collection (41%) and brand audit (8.3%) with 25% of teams sharing data with the community of practitioners and 12.5% sharing results with local policymakers. The analysis indicates debris is overwhelmingly composed of plastic from residential activities. The methods empower practitioners to collect and report on debris, ground-truthing global debris estimates, and illuminating the missing plastic problem.

**Keywords:** plastic; freshwater; marine; citizen science; India; policy; litter

## 1. Introduction

Several decades of research indicate the significant impact of marine litter to water, wildlife, ecosystems, and the economy [1–11]. South and East Asia are often described as prime contributors to the world's litter, due to dense populations living at the subsistence level, dependence on inexpensive single-use plastics, and little waste infrastructure [12–14]. India is of particular importance to the issue, as it has a coastline of nearly 7500 km, touching three seas and several major river systems, including the Ganga and Indus. Though Indian rivers are sacred both culturally and religiously, they are often littered. Based on their work cataloging plastics in the early 2000s in India, Sridhar et al. recommend further studies to pinpoint "quantity and quality of small plastic debris on intertidal zones" and a focus on "origin, transport and deposition" [15]. By 2016, Kumar and Sivakumar declared marine debris the "global problem least studied in India," and called for increased monitoring and evaluation [16]. This article introduces a replicable, scientific, inexpensive collection method for Indian marine debris research capacity building, and shares baseline data from ten cleanups to inform the problem of litter in India.

The authors held a week-long training workshop in Thiruvananthapuram, India in June 2019, with goals of augmenting local capacity for marine debris research; enriching empirical data; characterizing sources of debris; and sharing results with policymakers. This article reviews the literature on debris in India, outlines the workshop methods, and reports on data from ten collections in Kerala (4), Maharashtra (2), the Union Territory of Andaman and Nicobar Islands (1), Tamil Nadu (2), and the Union Territory of Lakshadweep (1); including categorization to determine sources, brand data, calculation of average density per square meter, Clean Coastal Index ranking, and completion rates for workshop participants.

Why promote ground-truthing macro debris estimates in developing countries? Researchers recommend focusing efforts on such communities, noting the importance of focusing on the world's most polluted rivers in countries characterized by swiftly developing economies and a lack of waste infrastructure [17]; that capture local deposition and intervene before it reaches the ocean [18]; that are positioned near high-density coastal regions [19]; and that concentrate on macro debris from consumer households [20]. In fact, Blettler and Wantzen describe the emphasis on microplastics in freshwater (imported from the developed world) as a form of scientific imperialism [20], whereas macroplastics should be of more concern in the developing world due to the lack of waste management. Supporting research on macro debris on Indian shores illuminates the story of waste in the developing world, establishing a baseline in local communities and informing policymakers, advocates, and practitioners.

## 2. Literature Review

There are many studies focusing on regional litter in Asia; such studies describe an increasingly uncontrollable situation [21] with high rates of micro and macro-plastic accumulation [22]; they detail collections of tens of thousands of pieces [23]; and reveal debris that has travelled from southeast Asia, south Asia, and Africa [23,24] even at times forcing fishermen to work against their best economic interests by avoiding preferred fishing sites [25]. Economists estimate marine litter to cost 1.28 billion USD per year (as calculated in 2008 dollars) across the 21 Asia-Pacific economies [26]. These costs, realized by the tourism, shipping, and fishing industries, stem from shoreline cleaning, fishing and shipping vessel damage tracked through insurance claim and repair data, and the removal of derelict fishing gear [26].

As a region with high population living at the subsistence level, heavy dependence on plastic packaging, and little waste infrastructure, Asia is often flagged as a prime contributor to the world's marine litter [12–14]. An Ocean Conservancy report names China, Indonesia, the Philippines, Thailand, and Vietnam as the greatest contributors of marine litter by volume [27]. Notably, data reveal that India contributes to marine debris in the top five, but the report creators highlight the east Asian countries as the top five with "geographic proximity" [27]. In 2016, South Asia inputs to waste accumulation totaled 334 million

tons, or an average 0.52 kg per person per day [28]. This number is expected to increase by 2025 [13] and double regionally by 2050 [28]. These conditions will be exacerbated by growing population trends and increasing development [28,29]. Blettler et al. note that many of the most polluted rivers can be found in Asia, yet a mere 14% of peer-reviewed studies stem from these important inland fisheries [17].

Here, we review the early literature on debris in India to establish context for this research. Studies on debris in India began with evidence in the early 1980s from Caranzalem Beach, Goa, on plastic pellets—also known as nurdles—ranging in density from 50–300 pieces/m$^2$ [30]. The author described a nearby Corlim Industrial Estate as the likely source, presciently noting " ... their non-degradable nature and continuous accumulation may prove to be an environmental hazard in future" [30]. The next studies on debris accumulation appeared in the early 2000s. Dharani et al. shared anecdotal evidence from Great Nicobar Island of the accumulation of substantial shoreline debris of non-local origin [31]. In evaluating the environmental pollution of the Alang-Sosiya shipyard in Gujarat, researchers found plastics represent 81.43 mg/kg in sediment samples, including "thermocol, Styrofoam, nylon, transparent plastics, colored plastics, and glass wool" attributed to shipbreaking [32]. Research of five sites in Karnataka revealed plastic abundance ranges from 6.9 to 37.9 g/m$^2$ by weight, recommending further studies, public education, plastic alternatives, and better disposal [15].

By the 2010s, the pace of research on debris in India increased rapidly. Duraisamy and Latha, working in Ennore port, Chennai, Tamil Nadu described anecdotal observations of "solid waste dumping [and] windblown debris," attributed to population, bank encroachment, and sewer discharge [33]. Ganesapandian et al., collecting debris over two years from beaches on the Gulf of Mannar, most frequently found plastic (48%), polystyrene (18%), and cloth (15%), attributing the litter to fishing, tourism, and sewage [34]. Kaladharan et al. sampled beaches, trawling hauls, and water over two years at eight sites in six Indian states finding "considerable quantities" of plastic ropes, pet bottles, sachets, milk covers and thin carry bags on beaches (0.145–9.8 g/m$^2$) and fishing grounds (32–85 g/haul) [35]. Describing the overwhelmingly negative consequences of sand mining, mangrove destruction, and plastic pollution on these coastal fishing areas, the authors recommend further study [35]. Jayasiri et al. studied four beaches in Mumbai over eleven months, most frequently finding plastic, with a mean abundance of 7.49 g and 68.83 items/m$^2$ in sediment samples and of 3.24 g and 11.6 items/m$^2$ for visible debris [36,37]. These researchers recorded significant variation across both time and space, attributing plastic contamination to "recreation, tourism, and religious activities" [37] and "consumer and household ... materials" as well as "fishing, boating, pharmaceuticals and manufacturing" [36]. Sampling monthly over two years from 2010–2012 at four sites in Karnataka, Sulochanan et al. most frequently found nylon and plastic ropes [38]. The mean density of reported debris was 233.86 $\pm$ 375.01 g/m$^2$ and 24.3 $\pm$ 25.5 items/m$^2$ (Thanneerbhavi), 141.7 $\pm$ 138.9 g/m$^2$ and 19.46 $\pm$ 15.57 items/m$^2$ (Panambur), and 420.11 $\pm$ 743.07 g/m$^2$ and 20.73 $\pm$ 18.72 items/m$^2$ (Chithrapur) [38]. The researchers recognized a relationship between abundance and proximity to discharge from the nearby Nethravathi and Gurupur rivers [38]. Working in Chennai, Veerasingam et al. sampled plastic nurdles along the high tide line, comparing pre- and post-flood levels [39]. The researchers found three times the number of pellets in the post-flooding sample (primarily polyethylene and polypropylene), attributable to influence of nearby rivers [39]. In research on nurdles at six sites in Goa, Veerasingam et al. reported polyethylene and polypropylene as the most abundant types, concluding southwest monsoons transport new micro plastic pellets to Goan beaches where they degrade [40]. Working on Marina beach, Chennai, Kumar et al. found 6872 individual pieces (129.7 kg) most of which was plastic (44.9%) including plastic bags, food wrappers and plastic cups [41]. They noted local recreation or land-based sources and recommended longer and larger-scale monitoring [41]. Kaladharan et al. evaluated 254 sites along all of eleven states of coastal India, determining plastics were the 'largest component" in their collections [42]. Fifty-one of the 254 beaches they surveyed were graded very clean (<1 g/m$^2$),

122 were rated clean (1.1–10 g/m$^2$), thirty-six were considered fair (10.1–20 g/m$^2$), seventeen were graded moderate (20.1–50 g/m$^2$), seven were rated littered (50.1–100 g/m$^2$) and twenty-one beaches were heavily littered (>100 g/m$^2$). The authors attributed debris levels to coastal urbanization, tourism, plastic packaging, and mobile phone use; they recommend education and legislation to combat the problem [42]. Evaluating microplastics in Vembanad Lake, Kerala, Sruthy and Ramasamy discovered microplastics in all of their sediment samples, calculating a mean abundance of 252.80 particles/m$^2$ with low-density polyethylene found most frequently and attributed to degradation of disposed items; the authors recommend prevention to alleviate the problem [43]. In the Gulf of Mannar, Vidyasakar et al. Noticed that their samples dominated by "polypropylene . . . followed by polyethylene, polystyrene, nylon and polyvinyl chloride," attributing this pollution to tourism and fishing [44]. Karthik et al., sampling microplastics at twenty-five sandy beaches across Tamil Nadu, found high tide line microplastic mean abundance was 1323 $\pm$ 1228 mg/m$^2$ compared to 178 $\pm$ 261 mg/m$^2$ at the low tide line; microplastics were found at highest density at beaches next to rivers, indicating land-based sources [45]. The most frequently found microplastics were polyethylene and polypropylene; authors recommend additional comprehensive studies that take into account human activity, processes, pathways, and seasonality [45]. Assessing macro and micro debris on Nallathanni Island, Gulf of Mannar, Krishnakumar et al. found plastic made up 73.2–100% of their samples, attributable to everyday consumer products (e.g., food, drink, health items) and fishing (e.g., nylon and polystyrene) [46]. Priya and Varunprasath surveyed 88 wetlands in Tiruppur district, Tamil Nadu, over ten months noting 44% of their sites had non-degradable waste (plastics) and 52% contained mixed waste including hazardous and radioactive waste [47]. To better conserve wetlands, they recommend public fora with representatives from education, research, and NGOs as well as locals [47]. Working in the Lakshadweep Archipelago, Joy et al. revealed contamination attributed to "anthropogenic pressure and developmental activities" including "diesel-based power generation, shipping activities, sewage sludge, plastic materials, fertilizers, construction, tourism activities, petroleum products, paints and pigments used in plastics, garbage and phosphate fertilizers" and noticed cadmium seriously threatens this reef ecosystem [48].

Research on debris in India seems likely to increase throughout the 2020s. Manickavasagam et al. working in South Juhu Creek, Mumbai quantified and analyzed debris flow through a channel, with mean results as 111 $\pm$ 5 pieces for high tide compared to 184 $\pm$ 12 pieces for low tide while the mean weights were 7.1305 $\pm$ 0.551 kg for high tide and 13.964 $\pm$ 1.234 kg for low tide [49]. Their work indicates a significant amount of material, mostly plastic, flows from high population areas through the channel to the sea, particularly at low tide and chiefly including macro and mega plastic waste [49]. Daniel, Thomas and Thomson collected data from six beaches in Kerala, finding most waste was plastic, amounting to 73.8% by count and 59.9% by weight [50]. The authors found the concentration of fishing-related plastic was four times greater on high intensity fishing beaches and that fishing-related plastic increased after monsoons; they recommend fishing community education and better collection of used and derelict fishing gear [50]. Assessing 21 islands of the Gulf of Mannar, Edward et al. revealed that, majority of the waste was abandoned fishing nets (43.17 $\pm$ 5.48%), damaging coral of the genera *Acropora* and *Montipora* [51]. They noted the critical role of reefs to the livelihoods of fishing communities, recommending management, debris reduction or elimination, monitoring, research, reduced fishing, gear maintenance, reef demarcation, outreach about ghost gear, removal and recycling of debris, education, aquaculture, artificial reefs, and solid waste management in nearby cities to alleviate the problem [51]. Focusing on abandoned, lost, or otherwise discarded fishing gear (ALDFG) along the length of the Ganga, Nelms et al. found 701 pieces of gear, including string (41%), net (40%), rope (10%), float (8%) and line (0.4%) for an average density of 0.013 ($\pm$0.038) items/m$^2$ [52]. The results indicated gear is not used for long; good disposal procedures do not exist; and regulations may be inadequate [52]. Furthermore, working along the length of the Ganga, Napper et al. found 140 microplastic

particles in 20 samples of ten sites, noting concentration was higher pre-monsoon and fibers were most prominent [53]. The researchers estimated that the Ganga, the Brahmaputra and the Meghna rivers may collectively release 1–3 billion particles into the Bay of Bengal daily [53].

While the methodologies employed vary greatly as determined by the goals of each study, this research indicates plastic pollution has grown as an environmental problem in Indian freshwater and coastal systems over time. As shown in the review, debris in India has been attributed to a wide range of sources and recommendations vary according to the study. The investigation of debris in India has become more frequent and analytical over the past four decades; and yet, these studies only skim the surface of the problem when considering the geographical scale of India and the importance researchers have placed on south Asia as a top polluting region.

The methods in this article provide baseline studies—it should be noted that without baseline studies, increasing waste degrades these ecologies without awareness of that which is being lost, an example of shifting baseline syndrome [54]. The methodology, when applied more broadly, can allow for more baseline studies along India's ample coastline, increasing and augmenting data on the types of debris, sources, and contextual management over time. In addition, such work highlights ways anthropogenic marine litter can be addressed by coupling citizen science and academic analysis, in this global waste hotspot.

## 3. Materials and Methods

Workshop goals were to train Indian participants (college professors, representatives of NGOs, and graduate students) in collection methods, in sorting and cataloging debris, in analyzing the collected material, and in writing up results in a policy brief. The workshop included a combination of lectures, instruction, and experiential methods. Participant capacity was increased by enhancing well-meaning beach cleanups that lack rigor and linking the results to policymaking. All participants took part in a cleanup, sorted, and analyzed debris by hand and created a policy brief. All participants were asked to return to their home communities around the country and complete a cleanup, to report those results to local policymakers, and to share the results and raw data on a ResearchGate project page. See Supplementary Materials for access to all open-source workshop materials.

Workshop participants were trained using methods modeled after the National Oceanic and Atmospheric Administration (NOAA) Marine Debris Shoreline methodology [54]. For riverside collections, researchers use survey flags to delineate an area along the river, marking off a 100-m swath of shoreline, 5 m deep (landward from the river shore) for a total collection area of 500 square meters (m$^2$). For coastal collections, at low tide researchers flag off a 100-m length of shoreline with depth from low to high tide lines. Total area varies according to the intertidal zone. After flagging the boundaries and creating lanes every 10 m perpendicular to the water, researchers move systematically within the lane, slowly walking and looking for, then collecting all debris, then turning and walking back down the lane, then turning up again, collecting everything visible within the given area attributable to humans until the full area has been walked. At times, groups might encounter very heavy or large items or materials that cannot or should not be moved. For example, during the Karamana River clean up event, we found several dozen funerary clay pots. Because of the proximity to the Parasurama Temple Thiruvallam, their role in holding cremated human remains, and that clay biodegrades, the team decided not to include this material in the count or remove them from the site.

The method combines elements of the NOAA *accumulation* [debris is removed from the whole shoreline at each visit to measure debris deposition over time] and *standing stock* [participants survey a 100-m-long stretch of beach to determine debris density but do not remove debris] [55]. Our method pulls from each to acquire baseline data about debris at a location while also removing debris. Unlike the accumulation method, we do not clear the entirety of a beach and then return periodically to measure accumulation over

time. Unlike the standing stock method, we *do* remove debris from the site. The proposed method focuses the survey on a 100-m length of shoreline, removing debris, and cataloging the material to understand its composition. The method is informed by its purpose: it is not about measuring accumulation rates over time or clearing a whole shoreline, but instead about establishing a baseline and the focused removal of debris that links scientific data to policy-making. By gathering data in this way, we widen the scope of baseline studies while building deep knowledge about the type and sources of debris. The resulting information is then used to inform policy.

For the locations included in this study, it is quite likely debris will vary over time due to seasons, the influence of monsoons, tourism, or local festivals. The goal of this project is not to understand the way debris accumulation is impacted by temporality [56]. Instead, the methods promoted in this workshop provide a snapshot of debris for local stakeholders that inform policymaking.

Researchers subdivided debris broadly (e.g., plastic, metal, glass), then into more specific categories (e.g., film, cans, foam), counting and weighing the material. The identifications and terms as well as categories proposed by NOAA methods were used. The NOAA methods do not mention microplastics, though they recommend only collecting items measuring over 2.5 cm. The NOAA method data collection sheets include plastic fragments (hard, foam, film) as well as fragments of metal and glass. The methodology described in this article does not pointedly collect and analyze microplastics (meaning, we do not collect sand, substrate, or water samples, do not sieve samples, do not analyze with Fourier-transform infrared spectroscopy). Debris and fragments of debris that can be easily collected by hand are included in the samples, meaning all those larger than one commonly accepted delineation for the definition of microplastics (i.e., 5 mm) [57]. After cataloging debris, it was assessed as a whole, by site, to understand if it might be attributable as storm debris, fishing gear, manufacturing material, shipping goods, and/or consumer waste.

We use the Clean Coast Index as a comparative tool to put the results into context. The Clean Coast Index provides a category designation that includes the categories very clean (0–0.1 parts/m$^2$), clean (0.1–0.25 parts/m$^2$), moderate (0.25–0.5 parts/m$^2$), dirty (0.5–1.0 parts/m$^2$), and extremely dirty (more than 1 part/m$^2$) [58]. The Clean Coast Index standardizes the cleanliness of beaches across sites globally.

A brand audit records brand information from collected materials to better understand origin and to hold manufacturers accountable [59,60]. Researchers completing audits made a note of every brand visible by item type (e.g., snack bag, drink container) and material (e.g., film plastic, hard plastic), then recorded total counts for each brand. After collections, these data were compiled, verified, and internet searches were used to determine the parent company for each brand. In some cases, there was not enough data to independently verify the manufacturer; only brands and parent companies that were independently verified are included in the results. In addition, workshop participants were given a policy brief template, shown example policy briefs, given general advice on engaging with policymakers, and wrote a policy brief during the workshop. This material on all stages of the study, results, community engagement, and policy recommendations can all be found in our opensource ResearchGate page, noted below in Supplementary Materials section.

## 4. Results

### 4.1. Debris Collection

Data collection took place between 19 March 2019 and 2 January 2020 and included ten collections at nine sites as shown in Figure 1.

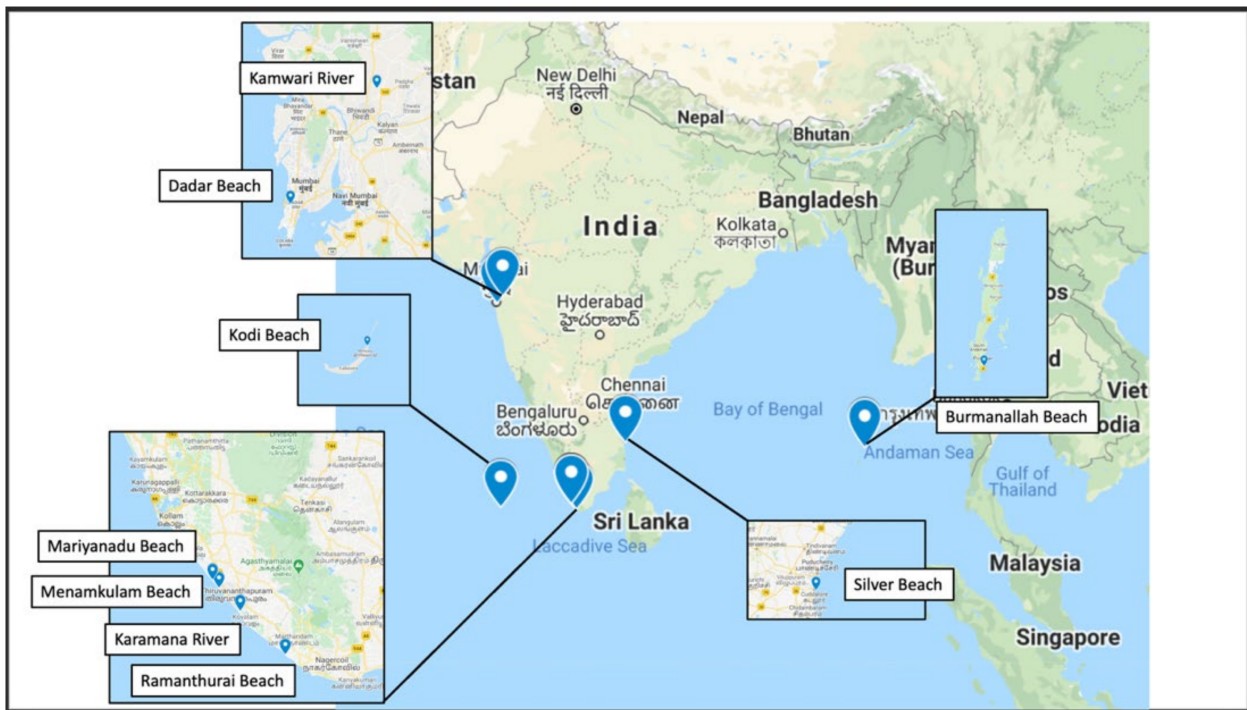

**Figure 1.** Research sites in India, from northwest, anti-clockwise: Kamwari River and Dadar Beach (Maharashtra), Kodi Beach (Union Territory of Lakshadweep), Mariyanadu Beach, Menamkulam Beach, Karamana River (Kerala), Ramanthurai Beach, Silver Beach (Tamil Nadu), and Burmanallah Beach (Union Territory Andaman and Nicobar Islands). (Image created using source material from Google Maps).

Across the ten collections, researchers collected over 33,000 pieces of debris weighing nearly 600 kg, 83.0% of which was plastic by count (57.3% plastic by weight) (Table 1). In all ten collections, plastic was the most frequently occurring type of debris, ranging from 45% (Ramanthurai Beach) to 89% (Menamkulam Beach, June) of the total sample. All debris counts and weight by site and subcategory of debris can be found in Appendix A.

**Table 1.** Cumulative debris tally and weight by material type from ten collections across India.

| Type of Material | Tally (%) | Weight in kg (%) |
|---|---|---|
| Plastic | 27,769 (83.0%) | 343.39 (57.3%) |
| Metal | 832 (2.49%) | 32.86 (1.94%) |
| Glass | 468 (1.40%) | 76.34 (5.48%) |
| Rubber | 154 (0.46%) | 11.64 (4.92%) |
| Processed trees | 1354 (4.04%) | 11.75 (3.99%) |
| Cloth, fabric, shoes | 1360 (4.06%) | 69.8 (12.7%) |
| Natural materials left by humans | 1105 (3.30%) | 23.89 (11.6%) |
| Mixed and other materials | 432 (1.29%) | 29.48 (1.96%) |
| Total | 33,474 | 599.15 |

Table 2 standardizes accumulation by providing the count and weight in context of the area of the site, then shares the associated rating according to the Clean Coast Index. For all sites, the material was overwhelmingly classifiable as consumer debris (i.e., not stemming from manufacturing, shipping, commercial or recreational fishing, or storms).

**Table 2.** Site area, total debris per site, and collection density with Clean Coast Index rating.

| Site | Date | Area (m²) | Tally | Weight | Debris Density by Count (Pieces/m²) | Clean Coast Index Rating | Debris Density by Weight (g/m²) |
|---|---|---|---|---|---|---|---|
| Menamkulam | 19 March 2019 | 2500 | 7420 | 43.5 | 2.97 | Extremely dirty | 17.4 |
| Menamkulam | 12 June 2019 | 2000 | 6653 | 106.5 | 3.32 | Extremely dirty | 53.25 |
| Karamana | 12 June 2019 | 500 | 1931 | 97.6 | 3.86 | Extremely dirty | 195.2 |
| Kodi Beach | 5 July 2019 | 2240 | 13,541 | 74.3 | 6.05 | Extremely dirty | 33.2 |
| Burmanallah Beach | 21 September 2019 | 5000 | 3135 | 325.10 | 0.63 | Dirty | 65.0 |
| Silver Beach | 21 September 2019 | 1500 | 2158 | 22.1 | 1.44 | Extremely dirty | 14.7 |
| Ramanthurai Beach | 23 September 2019 | 1000 | 381 | 17.9 | 0.38 | Moderate | 17.9 |
| Dadar Beach | 14 October 2019 | 700 | 519 | 5.55 | 0.74 | Dirty | 7.93 |
| Kamwari River | 24 October 2019 | 500 | 677 | 28.3 | 1.35 | Extremely dirty | 56.6 |
| Mariyanadu Beach | 2 January 2020 | 900 | 1488 | 6.23 | 1.65 | Extremely dirty | 6.92 |

*4.2. Brand Audit*

Six groups included brand audits in their assessment, yielding 2461 branded items across six sites (Burmanallah Beach, Menamkulam (2 collections), Kodi Beach, Karamana River, Mariyanadu Beach). An additional 427 items were submitted by the teams but did not have enough data to independently verify the manufacturer. Even with easily identifiable items, this at times becomes complicated as multinational and global brands may be manufactured by different groups depending on the country of sale. We made several allowances for these complexities. Brands produced by Hindustan Coca-Cola, were counted as American; 7-UP is bottled by PepsiCo outside of the United States (in the US it is bottled by Keurig/Dr Pepper) therefore we attributed it to PepsiCo in India. Finally, Oreo cookies are produced by Cadbury in India, whereas in the United States they are produced by Mondelez/Nabisco. For this assessment, we counted Oreo cookies as a Cadbury product (i.e., of British manufacture). When we discuss items and their country of origin, we do not presume that items have traveled from these places, as we have no evidence of this. Instead, evidence indicates that these items have been bought, sold, and consumed in India.

Our brand analysis indicates that the material found on Indian beaches is overwhelmingly produced by Indian companies (76%), with American companies ranking a distant second (13%). The branded material was also chiefly made of plastic (96%) and the majority of the material was used for packaging food and drink (93%), with 3% of the material made of glass, and less than 1% comprised of aluminum cans and tetra packs. The ten most frequently occurring brands within the sample account for 1538 pieces, or 62% of the material audited and are: Bisleri bottled water (16.9%), Indian Tobacco Company (ITC) (7.2%), Coca-Cola (5.8%), Maa Fruits (5.8%), the Milma milk cooperative (5.5%), PepsiCo (5.3%), Aryan Aqua India Pvt. Ltd. (5.0%), Parle Products (4.0%), Andaman and Nicobar Mineral Water (3.6%), and Haldiram Foods International (3.5%). The brand analysis confirms that a majority of Indian debris stems from local sources, which is important evidence when considering policymaking.

*4.3. Workshop*

The marine debris methods and practical workshop included 33 practitioners, representing 24 institutions, in five Indian states and two Union Territories. Of the 24 groups, ten (41%) conducted clean up events and two (8.3%) of these groups completed a brand audit in their home communities. Six (25%) of the teams collected and shared their data with the community of practitioners; three groups (12.5%) shared their results with politicians or policymakers; and one participant reported trying to meet with policymakers but being turned away. Some teams completed every element of the project (e.g., Kodi Beach, Minicoy Island [61]). Representatives from 14 (58%) groups did not complete a cleanup event upon their return home. The workshop included the host team of students and faculty from the University of Kerala, Department of Environmental Sciences, which conducted four

cleanups (two during the workshop in collaboration with the workshop participants). The University of Kerala team completed all aspects of the project for all of their cleanups and these data are included in the results, but as they were project partners rather than strictly workshop participants, these completions were not included in the measurement of the workshop's impact.

## 5. Discussion

While several workshop participants completed a cleanup (41%), the returns diminish for additional aspects of the methodology, including cataloging debris using the methods; conducting a brand audit; sharing the local results globally via ResearchGate; and reporting the results to policymakers. This is not surprising, as each component requires time and effort. The funding structure of future grants could include a stipend for project completion, which may serve as a motivator for busy people with many institutional responsibilities. Some participants chose not to complete the brand audit due to insufficient volunteer capacity or because the material they found was too degraded. This highlights an important point: that as debris ages, evidence disappears, and manufacturer accountability becomes more difficult.

No matter the country of context, policymakers and politicians may choose not to act on recommendations. We do not expect that policymakers will drastically change policy based on one outreach effort from the research workshop participants. Making links between science and policymaking may require repeated attempts and multiple forms of communication to engender long term policy change. That said, empowering stakeholders to collect scientific data and share the results with leaders strengthens civic engagement and allows communities to better understand and advocate for environmental protection.

It should also be noted that these efforts are not a commentary on whether India lacks a grassroots movement for evaluating and connecting data and policy on debris. There are many stakeholders working across India—whether as private citizens, as teachers, or through NGOs—to collect debris and advocate for policy change. This is simply one effort to infuse processes like these with scientifically replicable data and to build capacity to communicate results to policymakers. This work seeks to combat colonial or parachute science [62] and represents a collaborative effort between American and Indian researchers to build capacity in India.

To date, no changes in policy can be attributed to the action of workshop participants, but their work may contribute through rippling effects in communities over time. Arguably, local citizens concerned about an issue will have more sway and power to influence local manufacturers and distributors who may also be locals with a stake in the health of their environment. These scientific, replicable, and inexpensive baseline assessments, coupled with monitoring at representative locations over time, are key to tracking debris accumulation and the effectiveness of policy changes. Such baselines could be used by research teams to establish the problem and to write grants for studies that capture temporality and other aspects of the problem.

Plastic household waste was the main component of debris found in this study. In global south settings, both residential and industrial waste collection is irregular or non-functioning, and this waste often ends up dumped or lost in the environment [28]. Attributing the problem solely to mismanaged waste, however, fails to recognize the negative impacts of disposing of plastic in all forms. Alternatives to dumping include burning or burying, which are also detrimental to local environments, causing soil, groundwater, and air pollution. Throughout India, single-use plastics are ubiquitous, found as single serving packets of shampoo, coffee, hair color, and laundry detergent, sold in small corner 'penny' shops. The problem is made more complex when one considers that many consumers may not have the resources to buy or store larger containers of staple household products. As such, it is important that the problem is not framed as one of simply waste mismanagement. Single use plastics are used for a moment, but their environmental impact lasts many dozens or hundreds of years. Long-term solutions to the problem of waste in India and

other countries should include reducing single use plastics at the source, which will require stewarding a whole new system of no-waste practices and alternative materials to plastics.

The problem of river or marine debris is preventable—through banning single-use plastic items, improving waste infrastructure, incentivizing the refill and reuse economy, providing inexpensive biodegradable alternatives, and strengthening markets for materials that are recyclable (i.e., glass, some hard plastics). It is more efficient and cost-effective to remove pollution locally, rather than after it reaches freshwater and marine environments. It is even more efficient and cost-effective to prevent it from entering waterways altogether.

Addressing this issue in India will take a major shift on the part of the government, educational institutions, industry, community, and individuals. The Government of India has recently set a *Swacch Bharat* plan for a nationwide single-use plastic ban that could significantly impact the amount of single-use plastics found in the environment; yet the implementation of the plan has been stalled both at the national and state levels. The methods proposed here could be expanded by engaging higher education students and student volunteers of National Service Scheme (NSS, Indian Government sponsored service program conducted by Ministry of Youth Affairs and Sports) in these activities to generate awareness, data building, and advocacy.

Moreover, a consumer advocacy group surveyed 1936 businesses across Chennai in 2021 (two years after the initial single-use ban) and found that all establishments were using at least one form of banned plastics [63]. To use policy effectively for change, clear measures and enforcement should be adopted to curb usage of single-use plastics, and the costs of alternatives to plastics for the Indian marketplace must be considered; otherwise, there is likelihood that businesses will find ways to go around bans and/or substitute one single-use material with another. While the national government aimed to phase out single use plastics by 2022, the state of Kerala took a bold step in this direction by banning manufacture and sale of single use plastics from 1 January 2020; other states have made their own announcements for single-use reduction. For instance, Sikkim will ban all PET water bottles from the start of 2022, and Goa pledged to ban bags below 75 microns from September 2021. Essentially, it is up to states to decide how to implement these bans, and how to steward the transition to allow new practices to emerge and sustain—even in times of uncertainty. For instance, plastic reduction gains made before the global COVID-19 pandemic—such as recovery systems, and minimization of certain single-use items—were lost when workers were forced to stay at home and reusables were swapped for single-use items due to health and safety concerns.

While the community of practitioners created during this workshop have been stalled due to COVID-19, the group remains in contact and seeks opportunities to collaborate in the future to expand the program and continue to build capacity. Collaborations like this can infuse local policy with local data and may improve circumstances over time.

Plastic and other debris in India represent a significant threat to ecosystems, wildlife, and the economy. Training stakeholders in scientific, replicable, and inexpensive methods improves empirical data and empowers local stakeholders to better understand debris and share the results with policymakers. Expanding to implement this type of analysis in other countries can improve global data on marine litter, particularly in the developing countries often blamed with producing the most pollution.

**Supplementary Materials:** The following supporting information can be downloaded at: https://www.mdpi.com/article/10.3390/su14031928/s1. Material S1: How to conduct a beach cleanup, Material S2: How to conduct a river cleanup, Material S3: Blank data sheet, Material S4: Blank spreadsheet, Material S5: How to complete data sheets, Material S6: How to engage with local leaders, Material S7: How to write a policy brief, Material S8: Policy brief template, Material S9: Example policy brief, Connecticut, Material S10: Example policy brief, India

**Author Contributions:** Conceptualization, K.A.O. and J.D.S.; Data curation, K.A.O., J.D.S., K.C., S.K., A.B., N.V., M.S., S.A.V., A.J., V.H., R.P., P.J.K., D.K.K., P.M.M., S.C. and C.K.; Formal analysis, K.A.O., K.C., A.B., N.V. and M.S.; Funding acquisition, K.A.O. and J.D.S.; Investigation, K.A.O., J.D.S.,

K.C., S.K., A.B., N.V., M.S., S.A.V., A.J., V.H., R.P., P.J.K., D.K.K., P.M.M., S.C. and C.K.; Methodology, K.A.O., K.C. and A.B.; Project administration, K.A.O. and J.D.S.; Resources, J.D.S.; Supervision, J.D.S.; Visualization, K.A.O., N.V., S.A.V. and A.J.; Writing—original draft, K.A.O., J.D.S., K.C., S.K., A.B. and R.P.; Writing—review & editing, K.A.O., J.D.S., K.C., S.K., A.B., N.V., M.S., S.A.V., A.J., V.H., R.P., P.J.K., D.K.K., P.M.M., S.C. and C.K. All authors have read and agreed to the published version of the manuscript.

**Funding:** This research was funded by the Fulbright-Nehru Academic and Professional Excellence Awards program 8379-IN (K.A.O) and The National Geographic Society NGS-55326E-19 (K.A.O. and J.D.S).

**Institutional Review Board Statement:** The study was conducted in accordance with the Declaration of Helsinki, and approved by the Institutional Review Board of UNIVERSITY OF HARTFORD (PRO19050006 10 May 2019) for studies involving humans.

**Informed Consent Statement:** Informed consent was obtained from all subjects involved in the study.

**Data Availability Statement:** Data available on ResearchGate site: https://www.researchgate.net/publication/350189923_NEIEM_Data_File (Last Accessed 4 February 2022) DOI: 10.13140/RG.2.2.324-83.07207.

**Acknowledgments:** Authors gratefully acknowledge the assistance offered for this study by the Master of Science first semester students in the Department of Environmental Sciences of the University of Kerala, Thiruvananthapuram, India. We are grateful for the thorough and thoughtful comments of reviewers.

**Conflicts of Interest:** The authors declare no conflict of interest. The funders had no role in the design of the study; in the collection, analyses, or interpretation of data; in the writing of the manuscript, or in the decision to publish the results.

## Appendix A

**Table A1.** Debris by subcategory, across ten collections in India 2019–2020.

| | Menamkulam Beach March | | Menamkulam Beach June | | Karamana River | | Kodi Beach | | Burmanallah Beach | | Silver Beach | | Ramanthurai Beach | | Dadar Beach | | Kamwari River | | Mariyanadu Beach | |
|---|---|---|---|---|---|---|---|---|---|---|---|---|---|---|---|---|---|---|---|---|
| | Tally | Weight (kg) | Tally | Weight (kg) | Tally | Weight (kg) | Tally | Weight (kg) | Tally | Weight (kg) | Tally | Weight (kg) | Tally | Weight (kg) | Tally | Weight (kg) | Tally | Weight (kg) | Tally | Weight (kg) |
| PLASTIC | | | | | | | | | | | | | | | | | | | | |
| Hard plastic fragments | 564 | 0.90 | 206 | 26.41 | | | 3880 | 4.61 | | | 214 | 7.35 | 18 | 0.08 | 149 | 0.67 | 87 | 1.64 | 3 | 0.03 |
| Foam plastic fragments | 1593 | 0.55 | 2112 | 1.75 | 138 | 1.34 | 101 | 2.46 | 14 | 8.40 | 47 | 0.85 | 29 | 0.06 | | | 18 | 0.32 | 1060 | 0.42 |
| Film plastic fragments | 1838 | 1.75 | 1836 | 8.87 | 404 | 6.35 | 1848 | 9.27 | 281 | 7.00 | 263 | 0.70 | | | 56 | 0.18 | 11 | 0.14 | 22 | 0.03 |
| Food wrappers [1] | | | | | 133 | 3.63 | 90 | 0.37 | 725 | 30.80 | 7 | 0.06 | 8 | 0.02 | | | 33 | 0.08 | 6 | 0.02 |
| Beverage bottles | 38 | 1.75 | 217 | 10.91 | 191 | 5.77 | 32 | 0.66 | 912 | 49.80 | 2 | 0.01 | 13 | 0.37 | | | 11 | 1.28 | 17 | 0.65 |
| Other jugs or containers | | | 15 | 0.23 | 38 | 0.67 | 9 | 1.52 | 70 | 14.80 | 3 | 0.02 | | | | | 13 | 0.20 | | |
| Bottle or container caps | 203 | 0.80 | 353 | 2.40 | 151 | 0.31 | 351 | 0.91 | | | 2 | 0.00 | 19 | 0.05 | | | 20 | 0.15 | 10 | 0.03 |
| Cigar tips | | | 113 | 0.01 | | | | | | | | | 1 | 0.00 | | | | | | |
| Cigarettes | 183 | 0.25 | 45 | 0.16 | | | 13 | 0.00 | | | 1 | | | | | | | | | |
| Cigarette lighters | 2 | 0.02 | 5 | 0.08 | 3 | 0.04 | 1 | 0.01 | | | 1 | 0.00 | | | | | | | 1 | 0.15 |
| Bags [2] | | | 127 | 2.01 | 145 | 7.83 | | | | | 2 | 0.02 | 13 | 0.30 | | | 12 | 1.60 | 14 | 0.24 |
| Plastic rope and small net pieces | 778 | 0.55 | 185 | 2.93 | 8 | 0.08 | 1644 | 6.20 | 75 | 29.50 | 517 | 0.97 | 3 | 0.01 | | | 8 | 0.18 | 2 | 0.32 |
| Buoys and floats | 8 | 0.15 | 8 | 0.13 | | | | | 6 | 4.20 | 4 | 0.87 | 2 | 0.15 | | | 7 | 0.08 | | |
| Fishing lures and lines | | | 4 | 0.06 | | | | | 13 | 20.40 | 12 | 0.05 | 3 | 0.30 | | | 7 | 0.05 | | |
| Cups (including foamed plastics) | 16 | 0.25 | 74 | 1.17 | 20 | 11.33 | | | | | 1 | 0.00 | | | | | 18 | 0.08 | 2 | 0.03 |
| Plastic utensils | 16 | 0.02 | 3 | 0.05 | | | 8 | 0.02 | | | 6 | 0.03 | | | | | | | | |
| Straws | 30 | 0.02 | 113 | 1.25 | 9 | 0.041 | 187 | 0.05 | | | 3 | 0.01 | 7 | 0.00 | | | | | 1 | 0.00 |
| Balloons | 9 | 0.01 | 8 | 0.13 | | | | | | | | | 1 | 0.00 | | | | | 6 | 0.00 |
| Personal care products | 14 | 0.11 | 3 | 0.05 | 19 | 0.27 | 18 | 0.19 | 14 | 3.00 | | | 3 | 0.26 | | | 13 | 0.13 | 1 | 0.00 |
| Other [3] | 83 | 0.52 | 488 | 5.57 | 56 | 5.02 | 1857 | 10.98 | 65 | 2.00 | 3 | 0.01 | | | 113 | 0.54 | | | 21 | 0.05 |

**Table A1.** *Cont*.

| | Menamkulam Beach March | | Menamkulam Beach June | | Karamana River | | Kodi Beach | | Burmanallah Beach | | Silver Beach | | Ramanthurai Beach | | Dadar Beach | | Kamwari River | | Mariyanadu Beach | |
|---|---|---|---|---|---|---|---|---|---|---|---|---|---|---|---|---|---|---|---|---|
| | Tally | Weight (kg) | Tally | Weight (kg) | Tally | Weight (kg) | Tally | Weight (kg) | Tally | Weight (kg) | Tally | Weight (kg) | Tally | Weight (kg) | Tally | Weight (kg) | Tally | Weight (kg) | Tally | Weight (kg) |
| METAL | | | | | | | | | | | | | | | | | | | | |
| Aluminum/tin cans | | | 1 | 0.59 | | | | | 297 | 28.40 | 1 | 0.02 | | | | | | | | |
| Aerosol cans | | | | | 1 | 0.05 | | | | | | | | | | | | | | |
| Metal fragments | | | | | | | 51 | 1.06 | | | 2 | 0.02 | 19 | 0.01 | | | 6 | 0.12 | 2 | 0.02 |
| Aluminum foil | 44 | 0.02 | 57 | 0.04 | 4 | 0.01 | | | | | | | | | | | | | 1 | 0.00 |
| Other [4] | 59 | 0.25 | 111 | 1.75 | 173 | 0.49 | | | | | | | | | | | | | 3 | 0.01 |
| GLASS | | | | | | | | | | | | | | | | | | | | |
| Beverage bottles and jars | 48 | 13.85 | 49 | 12.23 | 24 | 12.37 | | | 90 | 26.50 | 10 | 1.45 | 13 | 3.50 | | | | | 2 | 0.85 |
| Glass fragments | | | 115 | 1.82 | 15 | | 19 | 0.38 | | | 1 | 0.01 | 5 | 0.01 | 17 | 0.44 | 17 | 0.17 | | |
| Other [5] | | | 11 | 0.17 | 3 | 0.094 | | | 29 | 2.50 | | | | | | | | | | |
| RUBBER | | | | | | | | | | | | | | | | | | | | |
| Gloves | 1 | 0.01 | 1 | 0.0005 | | | | | | | 1 | 0.01 | | | | | | | | |
| Rubber fragments | | | 10 | 9.36 | | | 25 | 0.70 | | | 15 | 0.03 | | | | | 6 | 0.11 | | |
| Rubber bands | 31 | 0.01 | 3 | 0.0012 | 8 | 0.003 | | | | | 4 | 0.01 | 2 | 0.00 | | | 6 | 0.02 | 5 | 0.00 |
| Other [6] | | | 7 | 0.13 | 1 | 0.005 | | | | | 1 | 0.00 | | | | | 27 | 1.25 | | |
| PROCESSED TREES | | | | | | | | | | | | | | | | | | | | |
| Cardboard | 517 | 0.70 | | | 1 | 0.06 | | | | | 2 | 0.01 | | | | | 3 | 0.55 | | |
| Paper [7] | | | 135 | 0.06 | | | 6 | 0.01 | | | 18 | 0.01 | 18 | 0.07 | | | 23 | 0.15 | 90 | 0.35 |
| Paper bags | | | | | | | | | | | 4 | 0.01 | | | | | 5 | 0.03 | | |
| Lumber/building materials | 161 | 0.55 | 103 | 1.63 | | | 31 | 1.56 | | | 0 | 0.00 | | | | | | | | |
| Popsicle sticks | | | 4 | 0.06 | | | | | | | 0 | 0.00 | | | | | | | | |
| Matchsticks [8] | | | 8 | 0.001 | | | | | | | 4 | 0.002 | 1 | 0.001 | | | | | | |
| Cigarette packets | 30 | 0.40 | 32 | 0.51 | 18 | 0.17 | 2 | 0.04 | | | 1 | 0.01 | | | | | | | | |

**Table A1.** *Cont.*

| | Menamkulam Beach March | | Menamkulam Beach June | | Karamana River | | Kodi Beach | | Burmanallah Beach | | Silver Beach | | Ramanthurai Beach | | Dadar Beach | | Kamwari River | | Mariyanadu Beach | |
|---|---|---|---|---|---|---|---|---|---|---|---|---|---|---|---|---|---|---|---|---|
| | Tally | Weight (kg) | Tally | Weight (kg) | Tally | Weight (kg) | Tally | Weight (kg) | Tally | Weight (kg) | Tally | Weight (kg) | Tally | Weight (kg) | Tally | Weight (kg) | Tally | Weight (kg) | Tally | Weight (kg) |
| Incense sticks | | | | | | | | | | | 7 | 0.01 | | | | | 22 | 0.02 | | |
| Other [9] | 89 | 0.15 | 1 | 3.92 | | | | | | | 3 | 0.01 | 15 | 0.71 | | | | | | |
| CLOTH, FABRIC, SHOES | | | | | | | | | | | | | | | | | | | | |
| Clothing | 41 | 2.85 | 15 | 0.24 | 25 | 7.84 | 272 | 4.98 | | | 4 | 0.96 | 9 | 0.12 | | | 11 | 3.80 | | |
| Shoes including flip flops | | | 54 | 9.71 | 19 | 2.55 | 36 | 5.72 | 64 | 20.20 | | | 3 | 0.66 | 1 | 0.22 | | | 5 | 0.46 |
| Towels or rags | | | | | 7 | 1.22 | 1 | 0.83 | | | 3 | 0.18 | | | | | 3 | 1.45 | | |
| Non-plastic rope or net pieces | | | | | | | 4 | 0.23 | | | 0 | 0.00 | | | | | 9 | 0.20 | | |
| Other [10] | | | 2 | 0.03 | 2 | 0.78 | 761 | 3.06 | | | 3 | 0.09 | | | | | 5 | 1.40 | 1 | 0.03 |
| NATURAL MATERIALS LEFT BY HUMANS | | | | | | | | | | | | | | | | | | | | |
| Ceremonial flowers | | | | | | | | | | | 210 | 1.45 | | | | | 30 | 0.13 | | |
| Herb bunches | | | | | | | | | | | 70 | 0.37 | 7 | 0.00 | | | 12 | 0.03 | | |
| Coconut | | | 1 | 0.02 | | | 411 | 15.42 | | | 8 | 0.25 | 15 | 2.86 | | | 3 | 0.08 | | |
| Coir | | | 1 | | | | | | | | 0 | 0.00 | | | | | 11 | 0.10 | | |
| Banana leaf | | | | | | | | | | | 4 | 0.02 | 6 | 0.17 | | | 2 | 1.50 | | |
| Other [11] | | | 8 | 0.05 | | | 105 | 0.70 | | | 159 | 0.68 | | | | | 42 | 0.07 | | |
| MIXED AND OTHER MATERIALS [12] | | | | | | | | | | | | | | | | | | | | |
| | 3 | 0.09 | 9 | 0.09 | 14 | 3.59 | 54 | 9.90 | 115 | 12.00 | | | 35 | 0.11 | 165 | 2.84 | 17 | 0.19 | 20 | 0.67 |

Gray shading denotes debris material broad categories [1] Menamkulam, included with film plastics. [2] Menamkulam March, included with film plastics. [3] Other plastic: Including toys, syringes, cement bags, lollipop sticks, plastic flowers, packaging, pens, diapers, sanitary pads, woven plastic bags, plastic wire, nylon, foam sponges). [4] Other metal: including metal caps, batteries, metal pins, washer, tin, keychain, blade. [5] Other glass: including medicine container, light bulbs. [6] Other rubber: including buoy, unrecognizable item. [7] Menamkulam March, paper included with cardboard. [8] Menamkulam March, matchsticks included with lumber. [9] Other rubber: including wooden pieces, playing cards, pencils. [10] Other cloth: including fabric pieces, hat, handbag. [11] Other natural materials: including food waste, fruit and vegetable peels, coconut shells, fish bones, charcoal, groundnut. [12] Mixed and other materials: including tetrapacks, fiberglass, electrical material, fused plastic and cloth, construction debris, paper food covers, wax cups, blister packs, helmet, capacitor, bag of refuse, polythene, sponge/rubber/thermacol, brick roof tile, poultry feathers, charcoal, silicon gel pack, fish traps with plastic cones and PET, football, gum, wipes.

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
