# Peer review of "Empowering Local Practitioners to Collect and Report on Anthropogenic Riverine and Marine Debris Using Inexpensive Methods in India"

_sustainability, doi:10.3390/su14031928_

Round 1

Reviewer 1 Report

This manuscript describes details of a project where methods of beach and river litter monitoring used by a North American institution (NOAA) were applied in various locations around India in partnership with local institutions. The main substance of the manuscript is an extensive literature review, description of the data gained from processing the litter, and discussion about how successfully the different parts of the methods were able to be implemented. There is no statistical analysis of the data, so this manuscript seems more like a description of an environment protection case study rather than scientific research, but I guess this still fits the scope for the Sustainability journal. I think other researchers and environmental project organisers will find it interesting to read about which parts of this project did/didn’t work, and the manuscript will be of modest usefulness for planning further related projects.

I have some minor comments:

line 25: change "take" to "took".

line 27: please clarify what exactly "replication among the community of practitioners" means, e.g. does this mean replication of sample size, replication of methods, or something else?

line 45: please check that this citation style fits the style of this journal – I have not seen a style such as [15:93] used in other articles I’ve reviewed for this journal.

line 69-70: I think the words "details" and "reveales" are meant to be "detail" and "reveal".

line 99: fix typo here

line 102: change to "increased"

line 106: change to "found". Please check the rest of the article for instances where present tense was used but which could be changed to past tense to be more clear.

line 168: only at this point I started realising the literature review for the introduction is exceptionally long. About here I was waiting to read about some aims and details about the current study, and was anticipating the winding-up of the review of results from past studies. Maybe you could add some section titles, including a "literature review" section, to the introduction. This way the reader will be aware that a different to usual format is used here (i.e. the literature review seems almost as important in the manuscript as the details of the project and data). Or possibly some of the literature review details could be moved to the discussion and related to the results from the current sampling.

line 231-235: Please clarify how your methods "combine the two" NOAA methods. It seems to me that as at most sites the litter was only collected once, this would correspond to the “standing stock” method.

line 321: I would say the sentence here is unnecessary and can be deleted.

line 333: change to "forms of communication"

line 366: I think it is unnecessary to mention the names of specific government ministers who are only very remotely related to an environmental study involving 10 small-scale litter clean-ups. For the same reason, I also do not think any mention of the COVID-19 pandemic is needed (line 371).

Line 374: Maybe do not use the concept of "training" so much, as I do not think too much training is really needed to just collect litter in a systematic way, categorise it, write about it to local politicians/policy-makers, and mention it on a website. Maybe change the term "training" to "incentivising", or something like that.

Author Response

Thank you for the helpful review of this manuscript—below please see your comments addressed one by one, with references to where we have amended the manuscript or an explanation of when we have not. Your time and effort are greatly appreciated.

This manuscript describes details of a project where methods of beach and river litter monitoring used by a North American institution (NOAA) were applied in various locations around India in partnership with local institutions. The main substance of the manuscript is an extensive literature review, description of the data gained from processing the litter, and discussion about how successfully the different parts of the methods were able to be implemented. There is no statistical analysis of the data, so this manuscript seems more like a description of an environment protection case study rather than scientific research, but I guess this still fits the scope for the Sustainability journal. I think other researchers and environmental project organisers will find it interesting to read about which parts of this project did/didn’t work, and the manuscript will be of modest usefulness for planning further related projects.

I have some minor comments:

line 25: change "take" to "took". corrected   

line 27: please clarify what exactly "replication among the community of practitioners" means, e.g. does this mean replication of sample size, replication of methods, or something else? Replication of the methods in other Indian regions - added this clarification to the abstract. 

line 45: please check that this citation style fits the style of this journal – I have not seen a style such as [15:93] used in other articles I’ve reviewed for this journal. We were using a colon in cases of direct quote, where [reference number: page number]. We have removed these instances to align with journal style. Through reading the style guide and examining published examples from the journal, we cannot find how references in this journal denote page number in cases of direct quote. If page numbers are needed please inform and we can include them.

line 69-70: I think the words "details" and "reveales" are meant to be "detail" and "reveal". corrected  

line 99: fix typo here corrected 

line 102: change to "increased" corrected

line 106: change to "found". corrected  

Please check the rest of the article for instances where present tense was used but which could be changed to past tense to be more clear. Corrected 

line 168: only at this point I started realizing the literature review for the introduction is exceptionally long. About here I was waiting to read about some aims and details about the current study, and was anticipating the winding-up of the review of results from past studies. Maybe you could add some section titles, including a "literature review" section, to the introduction. *Added this section, thank you for this insight for the flow of the paper and clarity of the subject matter. This way the reader will be aware that a different to usual format is used here (i.e. the literature review seems almost as important in the manuscript as the details of the project and data). We also noted the literature review in the abstract to better emphasize it. Or possibly some of the literature review details could be moved to the discussion and related to the results from the current sampling.

line 231-235: Please clarify how your methods "combine the two" NOAA methods. It seems to me that as at most sites the litter was only collected once, this would correspond to the “standing stock” method.

The standing stock method does not include the removal of debris, which is a critical component of this method. The accumulation method requires clearing the entirety of a shoreline and returning periodically to gauge accumulation over time. More details were added to the description to illuminate how we pull from each method and why.

line 321: I would say the sentence here is unnecessary and can be deleted: “To improve results, the training could expand to two weeks to allow more training, more experiential learning, and the development of greater skills and confidence for participants.” Deleted

line 333: change to "forms of communication" corrected

line 366: I think it is unnecessary to mention the names of specific government ministers who are only very remotely related to an environmental study involving 10 small-scale litter clean-ups. For the same reason, I also do not think any mention of the COVID-19 pandemic is needed (line 371).

We removed Modi’s name, though, respectfully, we believe COVID-19 is a relevant point to make considering how it thwarted many plastic-reduction policies and initiatives and has exponentially increased the use of single use plastics globally. 

Line 374: Maybe do not use the concept of "training" so much, as I do not think too much training is really needed to just collect litter in a systematic way, categorise it, write about it to local politicians/policy-makers, and mention it on a website. Maybe change the term "training" to "incentivising", or something like that.

While we appreciate this comment, we conceptualize this work as training. When someone learns to follow a methodology in a systematic way, training is required. Otherwise, the work is haphazard and non-scientific. The crux of this method is that people who may have been collecting debris in the past learn how to do it in a systematic, scientific, and replicable way. Their methods become more precise, and therefore are much more powerful and comparable across time, regions, and scales. It’s not about cleaning a beach to feel good—it’s about using local data to inform policy. When asked to go into the field to collect data, people will not follow a prescribed method unless trained to do so. In addition, creating reports for and communicating results to politicians and policy-makers is not a natural-born skill. Most people have no experience with sharing scientific results with policymakers to influence policy. Most people have not learned how to write a policy brief. They do not know how to craft a message in a way that appeals to the goals of politicians (i.e., connecting with a broad range of constituents, highlighting the economic costs and gains).  In our training, participants are taught how to communicate with politicians, how both the information shared and the way it is shared vary according to audience. We are not incentivizing them; we are training them to be better engaged and active citizens and advocates who link scientific knowledge with policy makers.

Reviewer 2 Report

This is both a very interesting paper that continues a long tradition of debris monitoring studies but in addition adds to the scope and innovativeness of mobilising measurement, involvement and reporting of community based investigation.

The opening review of the literature was comprehensive and conveyed a gravity to the worsening situation. The methods chosen and executed were carefully outline, the results in their fulness were placed in tables and discussion was succinct and to the point. The collection of brand data was a novel intervention. Likewise the training work shop anchors the research in community processes and ensures a standardisation of approach.

The paper ends on a pessimistic note, indicating that efforts to confront the societal problem of debris in rivers have been slow to gain political interest and even slower in gaining commitment as an ongoing social project. My question as a result is whether a sentence or two might be included in the conclusion to highlight any planned initiatives to enliven reportage of results.

Author Response

Thank you for the helpful review of this manuscript—below please see your comments addressed one by one, with references to where we have amended the manuscript or an explanation of when we have not. Your time and effort are greatly appreciated.

This is both a very interesting paper that continues a long tradition of debris monitoring studies but in addition adds to the scope and innovativeness of mobilising measurement, involvement and reporting of community based investigation.

The opening review of the literature was comprehensive and conveyed a gravity to the worsening situation. The methods chosen and executed were carefully outline, the results in their fulness were placed in tables and discussion was succinct and to the point. The collection of brand data was a novel intervention. Likewise the training work shop anchors the research in community processes and ensures a standardisation of approach.

The paper ends on a pessimistic note, indicating that efforts to confront the societal problem of debris in rivers have been slow to gain political interest and even slower in gaining commitment as an ongoing social project. My question as a result is whether a sentence or two might be included in the conclusion to highlight any planned initiatives to enliven reportage of results.

Sadly, we do end on a bit of a pessimistic note. In the discussion we have added the following:

Addressing this issue in India will take a major shift on the part of the government, educational institutions, industry, community, and individuals

The methods proposed here could be expanded by engaging higher education students and student volunteers of National Service Scheme (NSS, Indian Government sponsored service program conducted by Ministry of Youth Affairs and Sports) in these activities to generate awareness, data and advocacy.

And

While the community of practitioners created during this workshop have been stalled due to COVID-19, the group remains in contact and seeks opportunities to collaborate together in the future. Collaborations like this can infuse local policy with local data and may improve circumstances over time.

Reviewer 3 Report

This paper introduced a methodology to collect debris data and shared the collected baseline data from ten sites in India. This paper is written with a  good structure. It is easy to follow and enough details are provided to help understand the main idea. However, this paper lacks of novelty. The main methodologies are proposed by NOAA with no major optimizations. I agree that the work this paper introduced is meaningful, but this paper is more like a project report instead of a research paper. Some minor comments can be found below:

  • Line 102 - 194: introduced the related work in detail, but focused on the conclusions. Will it be better to introduce/compare/analyze their methodologies as well?
  • Line 252:  Will the debris data for a same location change over time? For example, there may be more people in the beach in the summer than the winter, so will the debris data  is higher in the summer than the winter?

Author Response

Thank you for the helpful review of this manuscript—below please see your comments addressed one by one, with references to where we have amended the manuscript or an explanation of when we have not. Your time and effort are greatly appreciated.

This paper introduced a methodology to collect debris data and shared the collected baseline data from ten sites in India. This paper is written with a  good structure. It is easy to follow and enough details are provided to help understand the main idea. However, this paper lacks of novelty. The main methodologies are proposed by NOAA with no major optimizations.

Please see additional commentary under the Methods section, paragraph three, that provide more detail about how we pull elements from two different NOAA methods and use them in a novel way.   I agree that the work this paper introduced is meaningful, but this paper is more like a project report instead of a research paper. Some minor comments can be found below:

  • Line 102 - 194: introduced the related work in detail, but focused on the conclusions. Will it be better to introduce/compare/analyze their methodologies as well?
    • We appreciate this comment-- we include an extensive literature review on debris studies in India because this has never been done before and we believe it provides context for the data we collect (both why studies of India are important and what the problem is like in India). While we are proposing a methodology, we are not evaluating and comparing these articles because of their methodology, but because of the focus on India. Including the information about methodologies for each of these articles would shift the focus and also place our manuscript well over the article length limits.
    • We feel what distinguishes the methods we propose is training stakeholders to collect data using replicable scientific methods (rather than well-meaning beach cleanups that lack rigor) as well as the link between this accumulated data and local policy-makers.
    • In response to this helpful comment, we added the following to the methods section:
      • The core concept was to build capacity by improving well-meaning beach cleanups that lack rigor and then to link the results to policymaking.
    • Line 252: Will the debris data for a same location change over time? For example, there may be more people in the beach in the summer than the winter, so will the debris data is higher in the summer than the winter?
      • Importantly, this is not an accumulation study. It is not about tracking debris over time and understanding how it changes seasonally. Instead, it is about training local stakeholders to collect and analyze debris data to inform local policymaking. We have added the following to the methods section to clarify:
      • For the locations included in this study, it is quite likely debris will vary over time due to seasons, the influence of monsoons, tourism, or local festivals. Little is understood about the way debris accumulation is impacted by temporality {Browne et al 2015} That said, the methods promoted in this workshop provide a snapshot of debris for local stakeholders that inform policymaking.

Thank you for this helpful feedback! 

Reviewer 4 Report

The manuscript presents a case study of litter collection from India, where the activities were carried out after training of people representative of local stakeholders.
Being submitted to a scientific journal, I expect the thread of the manuscript to be scientific questions, to be answered by the actions of the project and its outcomes. It does not seem the case. I am suggesting some changes in this respect. I hope to be able to read soon a revised version.
With respect to litter monitoring protocols:
. why after claiming a lack of common protocols, another protocol (modified after NOAA as stated in the text) was proposed?
. most detail on the protocol seems to relate to sites selection, also including river banks. What about items? Was the NOAA protocol applied in this case, and the same items IDs applied (for interoperability and replicability of the study)?
. if CCI was applied, why it only appears in the caption of table 2 and not in methods?
. there is a discussion about micro and macroplastics as study subject, but no reference for the size classes was adopted. There is a discussion about breakdown but in the protocol applied there is no mention of “fragments”, i.e. the fraction below the lower limit of 2.5 cm applied by most protocols, and above 5 mm (the most accepted upper size for microplastics, Frias, J. P. G. L., & Nash, R. (2019). Microplastics: finding a consensus on the definition. Marine pollution bulletin, 138, 145-147).
With respect to people engagement:
. does the discussion on science parachuting (I like this one Swanson, S. S., & Ardoin, N. M. (2021). Communities behind the lens: A review and critical analysis of Visual Participatory Methods in biodiversity conservation. Biological Conservation, 262, 109293. But I believe that specifically for plastics some paper should be out soon) relate to the investigation of why there is no grassroot movement to combat plastic pollution? Mind that, I fully agree with the increasing gap between scientists doing microplastics with increasingly costly techniques and NGOs doing macroplastics and “providing free data” to big repositories (Fanini, L., Defeo, O., Elliott, M., Paragkamian, S., Pinna, M., & Salvo, V. S. (2021). Coupling beach ecology and macroplastics litter studies: Current trends and the way ahead. Marine Pollution Bulletin, 173, 112951. for an analysis of publications on the two topics). I just would like to see this better developed.
. why is the CCI blamed to be developed for the exclusive use of the developed world? Does not serve the purpose of investigating the amount of litter present on beaches, wherever they are? Alkalay developed it to relate litter to the site of collection instead of amount of litter “per se”. In general, Chile has one of the best experiences on macroplastics litter studying and managing due to the social ecological work of Thiel and colleagues (Urbina, M. A., Luna‐Jorquera, G., Thiel, M., Acuña‐Ruz, T., Amenábar Cristi, M. A., Andrade, C., ... & Vargas, E. (2021). A country's response to tackling plastic pollution in aquatic ecosystems: The Chilean way. Aquatic Conservation: Marine and Freshwater Ecosystems, 31(2), 420-440.).
. the point about the process from collection to sorting and brand auditing to reporting is extremely interesting. I have myself noticed, many cleanups stop at “collection”, taking a picture with big sacks of litter, and that’s it. It would be great to further develop on this, and estimate ratios of bottom-up flows (also see my comment on scientific questions). But please also see Battisti, C., Poeta, G., Romiti, F., & Picciolo, L. (2020). Small environmental actions need of problem-solving approach: applying project management tools to beach litter clean-ups. Environments, 7(10), 87.

. I miss a section on current regulations and bans in the region. Usually those bans are evidence-based, on most common items found on beach collection. This is true, to my knowledge, in the EU but also in Morocco and some states of Brazil. These are all cases in which the quantitative analysis skipped science and went to policy-making.
. When discussing the narrowing percentages from collection to returning information, it should be considered that the concept of citizen science and monitoring also could result quite unfriendly to a non engaged public (Fanini, L., Marchetti, G. M., Serafeimidou, I., & Papadopoulou, O. (2021). The potential contribution of bloggers to change lifestyle and reduce plastic use and pollution: A small data approach. Marine Pollution Bulletin, 169, 112525). A shift to system thinking could be proposed.
. The introduction is very narrative; I would recommend drawing a timeline with the events reported, or a map.

Author Response

Thank you for the helpful review of this manuscript—below please see your comments addressed one by one, with references to where we have amended the manuscript or an explanation of when we have not. Your time and effort are greatly appreciated.

The manuscript presents a case study of litter collection from India, where the activities were carried out after training of people representative of local stakeholders.

Being submitted to a scientific journal, I expect the thread of the manuscript to be scientific questions, to be answered by the actions of the project and its outcomes. It does not seem the case. I am suggesting some changes in this respect. I hope to be able to read soon a revised version.

With respect to litter monitoring protocols: (Please let us know if we’ve misunderstood any of these questions about protocols)

Why after claiming a lack of common protocols, another protocol (modified after NOAA as stated in the text) was proposed?

We appreciate this comment-- we include an extensive literature review on debris studies in India because this has never been done before and we believe it provides context for the data we collect (both why studies of India are important and what the problem is like in India). While we are proposing a methodology, we are not evaluating and comparing these articles because of their methodology, but because of the focus on India. We note that the data collected so far in India include a wide range of protocols. The studies found in our literature review include anecdotal evidence, trawl net studies, as well as projects more like ours: beach or river coastline evaluations. Some of them use NOAA or similar methods.

We feel what distinguishes the methods we propose is training stakeholders to collect data using replicable scientific methods (rather than well-meaning beach cleanups that lack rigor) as well as the link between this accumulated data and local policy-makers.

In response to this helpful comment, we added the following to the methods section:

The core concept was to build capacity by improving well-meaning beach cleanups that lack rigor and then to link the results to policymaking.

Most detail on the protocol seems to relate to sites selection, also including river banks. What about items?

Items cannot be determined before the collection and analysis. The analysis follows the NOAA protocol for analyzing debris once collected, which is an international standard for the monitoring of marine debris. We have provided a link to the protocol in citation 54 (Opfer, S., Arthur, C., & Lippiatt, S. NOAA marine debris shoreline survey field guide. Available online: https://marinedebris.noaa.gov/noaa-marine-debris-shoreline-survey-field-guide (accessed on 1 Dec 2019). In Appendix A, you can see the full breakdown of material by type, location, weight and count. The analysis indicates debris is overwhelmingly composed of plastic and from local sources (we know local sources because of the brand audit and source of production, local brands vs international brands).

Was the NOAA protocol applied in this case, and the same items IDs applied (for interoperability and replicability of the study)?

YES, the NOAA protocol was applied for replicability of this study, as well as relevance for other national and international marine debris studies.

This sentence has been added: The identifications and terms as well as categories proposed by NOAA methods were used.

We also included more information in paragraphs three and four under the Methods section to better describe the choices we made and why.

If CCI was applied, why it only appears in the caption of table 2 and not in methods?

We have amended the methods section to include a reference to CCI.

There is a discussion about micro and macroplastics as study subject, but no reference for the size classes was adopted. There is a discussion about breakdown but in the protocol applied there is no mention of “fragments”, i.e. the fraction below the lower limit of 2.5 cm applied by most protocols, and above 5 mm (the most accepted upper size for microplastics,

This text was added to note how the NOAA methods address pieces smaller than 2.5 cm and also include the reference to a standard definition of microplastics. Thank you for these suggestions.

The NOAA methods do not mention microplastics, though they recommend only

collecting items measuring over 2.5 cm. The NOAA method data collection sheets include plastic fragments (hard, foam, film) as well as fragments of metal and glass. The methodology described in this article does not pointedly collect and analyze microplastics (meaning, we do not collect sand, substrate, or water samples, do not sieve samples, do not analyze with Fourier-transform infrared spectroscopy). Debris and fragments of debris that can be easily collected by hand are included in the samples, meaning all those larger than one commonly accepted delineation for the definition of microplastics (i.e., 5mm).

Please also see supplementary materials to the article for access to all open-source workshop materials.   

With respect to people engagement:

. does the discussion on science parachuting (I like this one Swanson, S. S., & Ardoin, N. M. (2021). Communities behind the lens: A review and critical analysis of Visual Participatory Methods in biodiversity conservation. Biological Conservation, 262, 109293. But I believe that specifically for plastics some paper should be out soon) relate to the investigation of why there is no grassroot movement to combat plastic pollution? Mind that, I fully agree with the increasing gap between scientists doing microplastics with increasingly costly techniques and NGOs doing macroplastics and “providing free data” to big repositories (Fanini, L., Defeo, O., Elliott, M., Paragkamian, S., Pinna, M., & Salvo, V. S. (2021). Coupling beach ecology and macroplastics litter studies: Current trends and the way ahead. Marine Pollution Bulletin, 173, 112951. for an analysis of publications on the two topics). I just would like to see this better developed.

Thank you for opening the discussion on parachuting. We are adamant anti-colonial-science researchers and have been greatly influenced by the work of fellow National Geographic Explorer Asha de Vos on this issue.  These studies were part of a week-long training workshop in Thiruvananthapuram, India in June 2019, with project goals of augmenting local capacity for marine debris research; enriching empirical data; characterizing sources of debris; and sharing results with policymakers. We have added this clarification in the second paragraph of the intro, as well as the abstract. The main objective was knowledge exchange and local capacity building for Indian stakeholders, so they could carry out these marine debris studies with their own students, in their own respective parts of the country, and catalyze more regular marine debris studies across the country (by Indian university professors).

Re: Micro/Macro and Scientist/NGO comment. Please also note the end of the intro mention of Bletter and Wantzen, describing the importation of the emphasis on micro studies as scientific imperialism. They advocate for more freshwater macro studies in developing countries. It is not that we think scientist should work on micro studies and that the ‘regular folks’ can do macro—it’s that we think it is important to empower people to infuse any collections with scientific rigor and then to communicate their political desires to local politicians.

Why is the CCI blamed to be developed for the exclusive use of the developed world? Does not serve the purpose of investigating the amount of litter present on beaches, wherever they are? Alkalay developed it to relate litter to the site of collection instead of amount of litter “per se”. In general, Chile has one of the best experiences on macroplastics litter studying and managing due to the social ecological work of Thiel and colleagues (Urbina, M. A., Luna‐Jorquera, G., Thiel, M., Acuña‐Ruz, T., Amenábar Cristi, M. A., Andrade, C., ... & Vargas, E. (2021). A country's response to tackling plastic pollution in aquatic ecosystems: The Chilean way. Aquatic Conservation: Marine and Freshwater Ecosystems, 31(2), 420-440.).  

We are familiar with Thiel and colleagues work (particularly his association with the Pew Charitable Trusts) and find them an important voice on social-ecological interdisciplinary work.

We simply note that CCI was devised in the developed world (not that it was created only for use in the developed world) and that the context and scale can be very different in the developing world, which often lacks waste infrastructure. We added a sentence to the methods that make clear why this is relevant.

We do think an important critique of the CCI would be its overemphasis on 'cleanliness,' at the expense of other considerations. For instance, one cant 'see' microplastics from a general eye-scan of the beach, let alone nanoplastics, or accumulated chemicals leaching from plastics. Cleanliness dialogues also do not go deep enough into the critique of increasing plastic use and production. Management and cleanliness are not sufficient to address the systems problem. But, we didn’t want to digress to far into this discussion within the paper—

We do find the CCI useful, which is why we used it to put the values into a broader context that might allow easy comparisons with other sites.

The point about the process from collection to sorting and brand auditing to reporting is extremely interesting. Thank you. I have myself noticed, many cleanups stop at “collection”, taking a picture with big sacks of litter, and that’s it. It would be great to further develop on this, and estimate ratios of bottom-up flows (also see my comment on scientific questions). But please also see Battisti, C., Poeta, G., Romiti, F., & Picciolo, L. (2020). Small environmental actions need of problem-solving approach: applying project management tools to beach litter clean-ups. Environments, 7(10), 87. Thank you, this is an interesting study. With the Brand Audit, we follow the protocols of Break Free From Plastic. They have been using brand auditing as a tool to address policymakers and keep businesses accountable. Annually they conduct a global brand audit   (https://www.breakfreefromplastic.org/brandaudit2021/). For the purposes of this paper and word count, we have not gone into all the detail, but brand audit was able to play the role for instance in advising the local authorities about which single-use items were most prevalent (for instance Milma milk sachets, the local milk brand).

I miss a section on current regulations and bans in the region. Usually those bans are evidence-based, on most common items found on beach collection. This is true, to my knowledge, in the EU but also in Morocco and some states of Brazil. These are all cases in which the quantitative analysis skipped science and went to policy-making.

We address gaps to policy near the end of the discussion section. We include marine debris studies from 10 different regions in India, thus if we included discussions on all the various policy this would turn into a policy paper rather than a marine debris methods and research paper. However, in the discussion we do mention the context and challenges with single-use plastics in the nation, and at the state level:
See paragraphs 7-9 in this section

When discussing the narrowing percentages from collection to returning information, it should be considered that the concept of citizen science and monitoring also could result quite unfriendly to a non-engaged public (Fanini, L., Marchetti, G. M., Serafeimidou, I., & Papadopoulou, O. (2021). The potential contribution of bloggers to change lifestyle and reduce plastic use and pollution: A small data approach. Marine Pollution Bulletin, 169, 112525). A shift to system thinking could be proposed.

This is an interesting study and can shed light on social media influence of scientific communication. This particular workshop did not lean on communication via social media. In India in particular, a face-to-face meeting with politicians and policymakers (the target audience) is much preferred, though this may change as a result of covid.

A systems thinking approach links with the brand audit - thinking about production, use and the bigger picture of plastic waste/pollution. As far as engagement, the science from the coastline and river surveys is translated into political advocacy (a non-engaged public could also be translated into non-engaged governance). The aim of this research is local science capacity building as well as providing tools to get local policymakers more engaged on marine debris issues. The general public was not the first audience for this research; although, results from for instance the brand audit could be used to share with the public about how their favorite brands are ending up on the beaches and in the waterways as pollution.

The introduction is very narrative; I would recommend drawing a timeline with the events reported, or a map.

We have separated the introduction and the lit review section to better clarify. The lit review gives the historical perspective of marine debris research in India.

Round 2

Reviewer 3 Report

Thanks for answering the questions. I did not think my major concerns were addressed: 1) I still feel this paper is more like a project report, not an innovative research paper. 2) I did not see enough new ideas in the methodology section. 3) as the results seem to be partial, the conclusions can not be used for the policy making.

Author Response

Please see attachment-- thank you. 

Reviewer 4 Report

The manuscript still reads like a hybrid between an activity report and a scientific contribution. I do believe that it is the author’s personal style, and I feel it is not fair to recommend to change it.
I do agree with the potential use of this paper as baseline, meaning that it could serve to provide data before some bans implementation -whether those bans stem from the workshop undertaken or not. It is worth stressing on this aspect though, e.g. at page 2 of the manuscript.

The concepts out of the paper from Bletter and Wanzen could be explicitated briefly, for those non familiar with the topic.

The cost of litter is in my opinion a powerful driver, also for non-motivated people. How would this connect to the engagement of policy-makers and managers? Are there local estimates?

To keep concepts clear and leave no option for confusion, I would avoid the mention of plastic resin pellets in introduction. They were not considered in the study after all. And also, their dynamics, from spillover to deposition and burial in the substrate, are way different than the macrolitter considered in your case. My recommendation is to state this from the beginning, showing awareness about all the fractions which can be considered, but focus directly (starting from the introduction) on the very same set of items targeted by your workshop.
A quick note on temporality. To my knowledge, the only way to deal with temporality is repetition, ideally at half of the pace of the phenomenon under study (e.g. two weeks for monthly phenomena, etc.). It is a huge demand, and I believe that not scientists nor volunteer citizens should do that alone; institutions would need to sustain them.

Still about CCI. I understand your statement. However, tourism is moving a pool of non-local people. It is their perception too that counts, and it is external to the local context. This should be considered.
On the same topic, I would finally stress that whatever the amount of litter present, if this is a threat to human welfare, this should trigger actions, as everyone deserves a clean environment, whether from developed or developing world.

Author Response

Please see attachment, thank you. 

Round 3

Reviewer 3 Report

The authors have addressed my questions. Looking forward to more creative contributions from the authors in this domain.